# Exosomal MicroRNAs as Epigenetic Biomarkers for Endometriosis: A Systematic Review and Bioinformatics Analysis

**DOI:** 10.3390/ijms26104564

**Published:** 2025-05-09

**Authors:** Cristina Maria de Araújo Medeiros Santos, Amaxsell Thiago Barros de Souza, Antonia Pereira Rosa Neta, Liziane Virginia Pereira Freire, Ayane Cristine Alves Sarmento, Kleyton Santos de Medeiros, André Ducati Luchessi, Ricardo Ney Cobucci, Ana Katherine Gonçalves, Janaina Cristiana de Oliveira Crispim

**Affiliations:** 1Postgraduate Program in Technological Development and Innovation in Medicines, Federal University of Rio Grande do Norte, Natal 59012-570, Brazil; cristinamsantos22@gmail.com; 2Postgraduate Program in Sciences Applied to Women’s Health, Federal University of Rio Grande do Norte, Natal 59012-310, Brazil; thiagoamaxsell@gmail.com (A.T.B.d.S.); liziaanev@gmail.com (L.V.P.F.); ricardo.cobucci.737@ufrn.edu.br (R.N.C.); ana.katherine.goncalves@ufrn.br (A.K.G.); 3Postgraduate Program in Health Sciences, Federal University of Rio Grande do Norte, Natal 59012-570, Brazil; antoniaprosaneta@gmail.com (A.P.R.N.);; 4Department of Clinical and Toxicological Analysis, Federal University of Rio Grande do Norte, Natal 59012-570, Brazil; ayanesarmento24@gmail.com; 5Research and Innovation Teaching Institute, League Against Cancer, Natal 59062-000, Brazil; kleyton_medeiros@hotmail.com; 6Postgraduate Program in Biotechnology, Potiguar University, Natal 59056-000, Brazil

**Keywords:** microRNAs, extracellular vesicles, exosomes, endometriosis, biomarkers

## Abstract

The clinical application of exosomal microRNAs as diagnostic biomarkers presents a promising approach for identifying potential markers of endometriosis. We conducted a systematic review of case–control studies to investigate exosomal microRNAs as epigenetic biomarkers potentially involved in the pathogenesis of endometriosis. A comprehensive literature search was performed across PubMed, Embase, Web of Science, and Scopus databases, yielding 702 studies, with 12 meeting the inclusion criteria after screening and full-text review. These studies included 191 women with confirmed endometriosis and 169 healthy controls. Quality assessment using the Newcastle–Ottawa Scale indicated a moderate quality across studies, with a common score of 5/9. In total, 668 exosomal microRNAs were found to be significantly differentially expressed between endometriosis patients and controls. In serum samples, 119 exosomal microRNAs were differentially expressed, with miR-22-3p, miR-320a, miR-320b, and miR-1273g-3p reported in more than one study. In endometrial tissue samples, miR-200c-3p and miR-425-5p were identified in more than one study, with miR-200c-3p consistently upregulated. Bioinformatic analysis indicated that these exosomal microRNAs are involved in key signaling pathways such as PI3K/Akt, MAPK, and TGF-β, which are associated with cell proliferation, migration, and inflammation. Despite these promising findings, variability in exosomal microRNA expression patterns across studies underscores the need for standardized methods and validation in large-scale, ethnically diverse cohorts. Future research should focus on rigorous validation studies to establish clinically relevant exosomal microRNAs for early diagnosis and improved patient outcomes.

## 1. Introduction

Endometriosis, defined by the growth of endometrial glands and stroma outside the uterine cavity, affects approximately 2–10% of women of reproductive age worldwide, totaling around 190 million women [1,2]. Early diagnosis of endometriosis remains challenging due to non-specific symptoms, including dysmenorrhea, dyspareunia, chronic pelvic pain, and infertility, compounded by the intra-abdominal location of endometriotic lesions, which often leads to diagnostic delays [3,4,5,6]. Although ultrasonography offers high sensitivity for detecting endometriotic cysts, its usefulness is limited in early-stage disease and requires histopathologic confirmation through invasive laparoscopic surgery [7,8]. Currently, no “gold standard” exists for the early diagnosis of endometriosis, highlighting the need for reliable, noninvasive biomarkers. Epigenetic biomarkers may enable earlier diagnosis and reduce the necessity for surgical intervention. Although several potential biomarkers have been explored, their sensitivity and specificity remain inadequate [9].

Evidence increasingly suggests a link between epigenetic factors and the pathogenesis of endometriosis [10]. Certain genetic loci reside in intergenic or intronic regions that, while non-coding, may still influence gene expression [11,12]. Exosomes have recently been shown to play a role in the pathophysiology of numerous diseases, including gynecological conditions [13]. These are small, membrane-bound vesicles (30–150 nm in diameter) secreted by various cell types. Exosomes contain a lipid bilayer that encapsulates functional molecules, including microRNAs (miRNAs), which participate in intercellular communication by influencing gene expression in target cells [14,15,16,17]. miRNAs regulate gene expression at the post-transcriptional level by binding to complementary messenger RNAs (mRNAs), leading to the repression of translation or degradation of target mRNAs [18]. They impact cellular processes such as apoptosis, differentiation, and proliferation and are implicated in various diseases, including disorders affecting female reproductive function [19,20,21]. However, the role of exosomal miRNAs in endometriosis remains underexplored and is not yet well understood.

A recent systematic review involving 1527 women with endometriosis reported significant dysregulation of several circulating miRNAs in serum and plasma, including miR-17-5p, miR-451a, let-7b-5p, miR-20a-5p, miR-143-3p, miR-199a-5p, and miR-3613-5p [22]. However, circulating miRNAs in the bloodstream are vulnerable to degradation by endogenous RNases. In contrast, exosomal miRNAs are stable in the blood and generally resistant to ribonuclease activity, making them promising candidates for disease biomarkers [17]. We hypothesized that exosomal miRNAs could serve as robust phenotypic signatures for the noninvasive diagnosis of endometriosis. Therefore, this study aimed to evaluate exosomal miRNAs as potential epigenetic biomarkers involved in the pathogenesis of endometriosis.

## 2. Results

### 2.1. Study Selection

The initial database search identified a total of 702 studies. After removing 119 duplicates, 583 studies remained for the title and abstract screening, from which 567 were excluded. Based on the inclusion criteria, 15 studies were selected for full-text review. Upon assessing eligibility, two studies were excluded due to incomplete data, and one because it evaluated other extracellular vesicles without separately reporting exosome findings. Ultimately, 12 studies [23,24,25,26,27,28,29,30,31,32,33,34] met the study criteria and were included in the pooled synthesis. The study selection process is illustrated in the PRISMA flowchart (Figure 1).

### 2.2. Study Characteristics

The characteristics of the 12 included studies are presented in Table 1. These studies, published between 2016 and 2024, involved a total of 191 women with confirmed endometriosis from China [24,25,26,27,28,29,30,31,32,33,34] and the USA [23]. Diagnoses were established through magnetic resonance imaging (MRI) and/or laparoscopy, with histological confirmation when possible. Most participants presented with moderate-to-severe endometriosis (stages III–IV) [26,28]. Seven studies did not specify the stage of endometriosis [23,24,25,28,31,32,33]. One study included only women with stages II–IV [34], while another focused exclusively on stage III [30]. Additionally, the studies involved 169 control participants, all of whom were healthy women.

Most studies included patients who had not received hormonal treatment in the past 3 months [23,24,27,29,32,33,34]. Two studies excluded patients who had taken hormonal treatment within the last 6 months [26,28]. Only one study did not specify the hormonal treatment status of the participants [23], and three did not report the length of the period during which participants had refrained from hormonal use [25,30,31].

Six studies did not specify the menstrual phase of participants [23,25,26,28,29,33]. One study differentiated between the proliferative and secretory phases [27]. Three studies included only women in the secretory phase [23,32,34], while two studies [30,31] collected samples exclusively in the proliferative phase. Eight studies analyzed eutopic and/or ectopic endometrial tissues [23,25,26,28,30,31,33,34], while three used serum samples [24,27,29]. Exosomes were detected using nanoparticle tracking, transmission electron microscopy, and Western blotting, with ultracentrifugation [23,26,27,29,30,31,32,33] being the primary isolation method. RT-PCR was the primary method for miRNA detection.

### 2.3. Quality Assessment

The quality of the included studies was evaluated using the Newcastle–Ottawa scale (Table 2), with the most common overall quality score being 5 out of 9. The primary limitations of these studies included insufficient comparability between cases and controls, along with inconsistent definitions of control groups.

### 2.4. miRNA Dysregulation in Endometriosis

All included studies analyzed dysregulated miRNAs, identifying those with the highest potential, which were subsequently validated. Across all studies, 668 exosomal miRNAs were found to be significantly differentially expressed between endometriosis patients and controls. Table 3 presents the dysregulated miRNAs identified in these studies.

#### 2.4.1. Serum

A total of 119 exosomal miRNAs were found to be differentially expressed in the serum of endometriosis patients (Table 2). Only three studies investigated exosomal miRNAs in serum [25,27,29]. Four miRNAs (miR-22-3p, miR-320a, miR-320b, and miR-1273g-3p) were differentially expressed in more than one study. Specifically, miR-22-3p, miR-320a, and miR-320b were identified in both studies by Huang et al. [24] and Zhang et al. [29] but displayed opposing trends, with upregulation reported by Zhang et al. [29] and downregulation by Huang et al. [24]. Additionally, miR-1273g-3p was identified in two studies [24,27], with downregulation reported by Huang et al. [24] and upregulation by Wu et al. [27].

#### 2.4.2. Endometrial Tissues

The systematic review of exosomal miRNA expression profiles in endometrial tissues revealed several miRNAs with consistent regulation patterns across eight studies [23,25,26,28,30,31,33,34]. A total of 308 exosomal miRNAs were found to be differentially expressed. Among these, miR-200c-3p and miR-425-5p were reported in more than one study. miR-425-5p was upregulated in the work of Zhang et al. [28] and downregulated in the work of Zhou et al. [34]. Interestingly, miR-200c-3p was the only miRNA consistently upregulated across studies, as reported by Zhou et al. [34] and Jiang et al. [25].

#### 2.4.3. Tubal Fluid

In tubal fluid analysis, 14 exosomal miRNAs were found to be differentially expressed, as reported by Zhang et al. [32], who applied a fold-change threshold greater than 2 and a *p*-value < 0.05. miR-6087, miR-4443, miR-5194, and miR-6834-3p were significantly upregulated, indicating potential roles in the pathophysiology of fallopian tube conditions. Conversely, miR-6747-5p, miR-1273f, miR-5699-5p, miR-10b-3p, miR-3911, miR-4419a, miR-4441, miR-4655-3p, miR-6778-5p, and miR-6845-5p were downregulated, suggesting reduced expression in the tubal fluid of women with endometriosis. Interestingly, miR-10b-3p was downregulated in the tubal fluid by Zhang et al. [32] but was reported as upregulated in endometrial tissue by Zhou et al. [34].

#### 2.4.4. Uterine Aspirate Fluid

The analysis of uterine aspirate fluid by Jiang et al. [25] identified nine miRNAs with differential expression based on a fold-change threshold of ≥1.50 and a significance level of *p* < 0.05. Upregulated miRNAs included miR-210-3p, miR-20b-5p, miR-625-5p, miR-342-5p, miR-155-5p, miR-146a-5p, and miR-130b-3p, while miR-335-3p and miR-132-5p were downregulated. Jiang et al. [25] also reported miR-210-3p as being upregulated in both uterine aspirate fluid and endometrial tissue.

#### 2.4.5. Leukorrhea

Only Zheng et al. [33] investigated exosomal miRNAs in leukorrhea, identifying 217 differentially expressed miRNAs (*p* < 0.05). Among these, miR-202-3p and miR-202-5p were upregulated. These miRNAs were also reported as upregulated in endometrial tissue by the same author.

### 2.5. Bioinformatic Analysis

A total of 9 selected miRNAs and 1644 target genes were used to construct a miRNA–mRNA interaction network (Figure 2). Gene Ontology (GO) Biological Process enrichment analysis provided insights into the molecular mechanisms involving these molecules (Figure 3). The target genes associated with the selected miRNAs were primarily involved in processes such as transcription regulation by RNA polymerase II, gene expression regulation, transcription from DNA templates, cell population proliferation, protein phosphorylation and modification, positive regulation of transcription and gene expression, nervous system development, and negative regulation of translation, transcription, and cell motility.

The Kyoto Encyclopedia of Genes and Genomes (KEGG) enrichment analysis revealed that the target genes of the miRNAs were primarily associated with the following pathways: long-term potentiation, Hedgehog signaling, renal cell carcinoma, microRNAs in cancer, TGF-β signaling, longevity regulation, endocrine resistance, cellular senescence, axon guidance, oocyte meiosis, thyroid hormone signaling, MAPK signaling, proteoglycans in cancer, focal adhesion, Rap1 signaling, human cytomegalovirus infection, PI3K−Akt signaling, actin cytoskeleton regulation, endocytosis, and general cancer pathways. The KEGG enrichment results are illustrated in Figure 4.

## 3. Discussion

The clinical potential of miRNAs as diagnostic biomarkers opens promising avenues for detecting disease, given their pivotal role in gene transcription and post-transcriptional regulation. Exosomal miRNAs, protected within extracellular vesicles, exhibit greater stability and targeted delivery compared to free-circulating miRNAs. This enhanced stability and specificity contribute to their potential predominance in various biological contexts [35].

This study systematically reviews exosomes in diverse tissues of endometriosis patients, marking the first comprehensive analysis of circulating exosomal miRNAs in 191 women. The findings provide a foundation for understanding the diagnostic relevance of miRNAs in endometriosis.

Our analysis identified 119 exosomal miRNAs with differential expression in serum samples from endometriosis patients compared to controls. Notably, only four miRNAs—miR-22-3p, miR-320a, miR-320b, and miR-1273g-3p—showed differential expression in more than one study, although their regulatory patterns varied, complicating the determination of their roles in endometriosis pathogenesis.

Growing evidence implicates these miRNAs in key endometriosis-related pathways, such as IKKβ/NF-κB, PI3K/Akt/mTOR, and MAPK (ERK1/2, p38, and JNK). These pathways are involved in cellular proliferation, migration, invasion, and apoptosis, essential processes that become dysregulated in endometriosis. Interestingly, KEGG pathway enrichment analysis highlighted the MAPK, PI3K/Akt, and TGF-β signaling pathways as significantly enriched in the miRNA targets [36].

The PI3K/Akt pathway, a prominent focus in endometriosis research, underscores a pathogenic role in the disease’s progression [36,37]. PI3K activation triggers Akt activation, with studies showing PI3K/Akt pathway overactivity and PTEN gene downregulation in eutopic endometrium, which may facilitate endometriotic cell adhesion and proliferation at ectopic sites [38,39]. Research on exosomal miR-301a-3p has shown its mediation of macrophage polarization through the PTEN–PI3K axis [40], while miR-146a-5p in exosomes from ectopic endometrial stromal cells has been linked to macrophage polarization through TRAF6 targeting [41].

In addition, miR-22-3p, a notable regulator of DNA-templated transcription, has been shown to inhibit the PTEN/PI3K/Akt pathway, associated with endometriosis cell proliferation [42]. Downregulation of miR-22-3p observed in patients may lessen its inhibitory effects on targets like HIF-1α, promoting hypoxia, inflammation, angiogenesis, and endometrial proliferation—all central to endometriosis pathogenesis [43,44]. Zhang et al. also demonstrated that exosomal miR-22-3p can suppress NF-κB signaling via SIRT1, potentially mitigating inflammation in ectopic endometrial tissue [28].

The miR-320 family, similarly, implicated in endometriosis, has shown involvement in cell proliferation and invasion, acting through pathways such as PI3K/Akt and STAT3 [45]. Studies suggest that miR-320 downregulates Wnt/β-catenin and TGF-β1, processes heightened in endometriosis, thereby potentially providing a protective effect [46,47,48].

Interestingly, inconsistent regulatory patterns emerged for miR-22-3p, miR-320a, miR-320b, and miR-1273g-3p across studies, challenging their immediate use as consistent biomarkers. Despite this, these miRNAs demonstrate strong potential as candidates for diagnostic panels, pending further cross-validation.

Among miRNAs in endometrial tissue, only miR-425-5p and miR-200c-3p were consistently identified. The miR-425-5p, associated with MAPK pathway regulation [49], presents conflicting regulation across studies, while miR-200c-3p was upregulated in two studies, aligning with its known tumor-suppressive function in endometriosis [50,51,52,53].

Finally, exosomal miR-210-3p was found upregulated in both uterine aspirate fluid and endometrial tissue, suggesting an interesting locus biomarker for endometriosis. Our GO analysis highlighted miR-210-3p’s involvement in transcription regulation, marking it as a potential biomarker for endometriosis. Studies indicate its role in DNA repair and hypoxia response [54], pathways central to endometriotic tissue progression.

This systematic review has limitations that should be considered. The heterogeneity among included studies in terms of sample size and endometriosis stages may affect the comparability of results. Additionally, the geographic and ethnic diversity of study populations may influence the expression profiles of miRNAs, limiting the generalizability of findings. Finally, although we identified promising exosomal miRNAs, the validation of these in independent large-scale cohorts is necessary to confirm their diagnostic accuracy and clinical utility.

## 4. Materials and Methods

### 4.1. Protocol and Registration

This systematic review followed the guidelines of the Cochrane Handbook and the Preferred Reporting Items for Systematic Reviews and Meta-Analysis (PRISMA) [55,56]. The protocol for this meta-analysis was registered in the International Prospective Register of Systematic Reviews (PROSPERO) database (registration number: CRD42024532794). The PRISMA checklist is given in Appendix A Appendix A.

### 4.2. Eligibility Criteria

The inclusion criteria for this review comprised published articles meeting the following: I. Studies involving women diagnosed with endometriosis; II. Studies evaluating the differential expression and regulation pattern (up- or downregulation) of exosomal miRNAs using microarray and/or RT-PCR in various human samples, including endometrial tissue, tubal fluid, leukorrhea, uterine aspirate fluid, and serum. The research question, structured in the PECOS format, was “Could exosomal microRNAs serve as potential epigenetic biomarkers for diagnosing endometriosis?” with criteria as follows:Participants: Women clinically diagnosed with endometriosis;Exposure: Exosomal miRNAs;Control: Healthy women;Outcome: Expression and regulation patterns of exosomal miRNAs in endometriosis patients, along with downstream pathways;Study type: Case–control studies.

Exclusion criteria included studies involving animal models, those based exclusively on cell cultures, brief communications, other types of analyses on the topic, and articles behind paywalls unavailable online. There were no restrictions on publication date, language, or sample type. Full texts of all eligible studies were sought for inclusion.

### 4.3. Search Strategy

We searched PubMed, Embase, Web of Science, and Scopus databases. An experienced librarian developed and piloted a search strategy for PubMed, using MeSH terms for “Exosomes”, “Endometriosis”, and “MicroRNAs”, which was adapted for other databases. Searches were conducted up to July 2024 without language restriction. References from included studies were reviewed to identify additional relevant papers. All strategies are provided in Appendix A Appendix A.

### 4.4. Selecting Studies

Results from multiple databases were imported into Rayyan software (https://www.rayyan.ai/, accessed on 1 November 2024), where duplicates were removed. Two investigators (CMAMS and LVPF) performed pilot screenings, resolving disagreements through calibration meetings to refine inclusion criteria. After this, studies were independently screened by title and abstract, followed by a review of full texts for relevance. Discrepancies were resolved by consensus or with a fourth investigator (ATBS).

### 4.5. Data Extraction Process

Two independent investigators (CMAMS and ATBS) extracted data chronologically, with discrepancies resolved through discussion or assistance from a third investigator (JCOCF). Data were organized into a standardized Excel form (Microsoft Corporation, Redmond, WA, USA), capturing the first author, year, country, sample size, menstrual cycle phase, endometriosis stage, diagnostic criteria, sample type, miRNA and exosome detection methods, and exosome isolation method. A second form documented the following: I. Number of differentially expressed exosomal miRNAs, II. Criteria for differential miRNA expression, and III. Dysregulated exosomal miRNAs. If further information was needed, corresponding authors were contacted by email.

### 4.6. Quality Evaluation

Quality was assessed independently by two investigators (CMAMS and LVPF) using the Newcastle–Ottawa Quality Assessment Scale (NOS) [57], which is validated for non-randomized studies. Disagreements were resolved by consensus with a third investigator (ATBS).

### 4.7. Bioinformatic Assessment

A bioinformatic analysis was conducted to explore molecular interactions between miRNAs and endometriosis. Only miRNAs identified in more than one study were included. Target prediction was performed using the miRDIP v5.3 database with a score threshold of “Very High” (top 1%) [58]. The miRNA–mRNA interaction network was visualized with Gephi (version 0.10.1). Gene Ontology (GO) Biological Process enrichment analysis was conducted using EnrichR (https://maayanlab.cloud/Enrichr/, accessed on 1 November 2024), focusing on the top 100 terms with a *p*-value < 0.05 [59,60]. Terms were further analyzed if shared by at least five miRNAs studied. Two of the initially identified 11 miRNAs were excluded due to a lack of target data. KEGG pathway analysis and visualizations were performed with ShinyGO v0.8 [61], and dot plots were created in RStudio (version 2023.06.0+421) using the ggplot2 package (version 3.4.2). The list of target genes is provided in Appendix A Appendix A.

## 5. Conclusions

This systematic review highlights the potential of exosomal miRNAs as biomarkers in endometriosis, though current findings underscore the necessity for further validation. While select miRNAs demonstrate promise, their application as clinical biomarkers is not yet feasible due to issues of specificity and sensitivity. Variability among studies, including differences in sample size, endometriosis staging, and ethnic diversity, emphasizes the need for standardized procedures and validation in large, diverse cohorts.

Ultimately, while exosomal miRNAs offer a window into the complex pathophysiology of endometriosis, large-scale, standardized studies are essential to confirm these findings and enable the use of miRNAs in early diagnosis and targeted treatments for endometriosis.

## Figures and Tables

**Figure 1 ijms-26-04564-f001:**
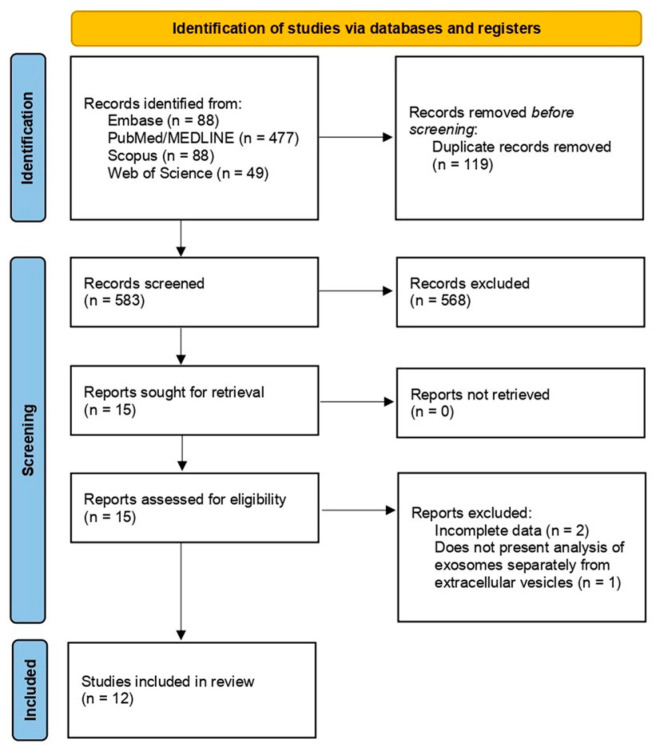
PRISMA flowchart.

**Figure 2 ijms-26-04564-f002:**
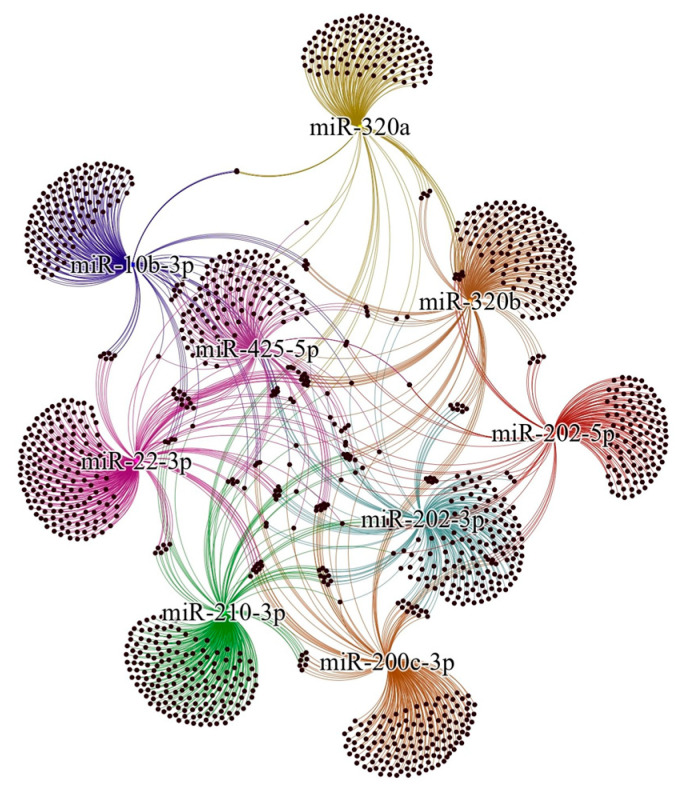
miRNA–mRNA network analysis of miRNAs possibly involved in endometriosis. Colored nodes represent miRNAs and black nodes are their target genes. miRNA: microRNA; mRNA: messenger RNA.

**Figure 3 ijms-26-04564-f003:**
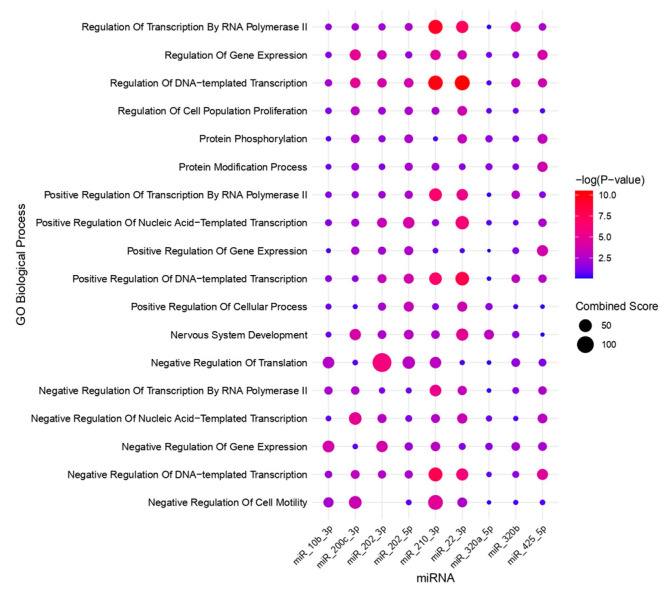
Gene Ontology (GO) enrichment analysis for microRNA targets in endometriosis, where the color of the dots indicates the *p*-value, with red representing more significant values (lower *p*-values) and blue representing less significant values (higher *p*-values). The size of the dots corresponds to the Combined Score (CS), with larger dots indicating a higher CS.

**Figure 4 ijms-26-04564-f004:**
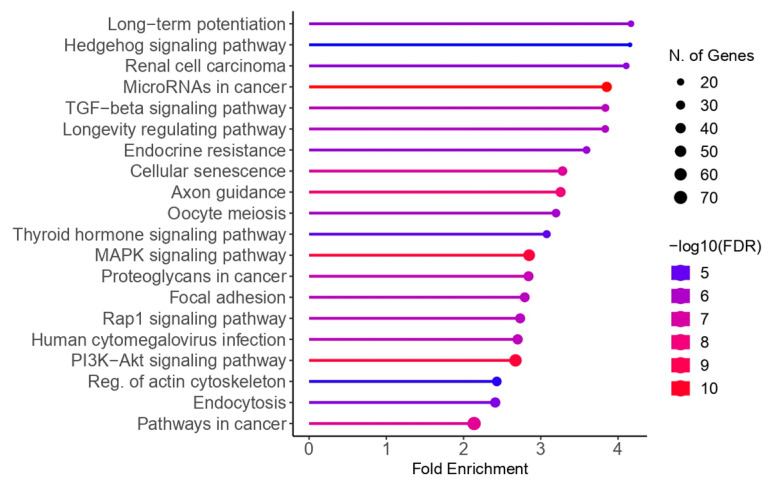
Kyoto Encyclopedia of Genes and Genomes (KEGG) enrichment analysis for microRNA targets in endometriosis. The color of the graph indicates FDR in a −log10 function, with red representing more significant values and blue representing less significant values. The size of the dots corresponds to the number of genes involved in each pathway.

**Table 1 ijms-26-04564-t001:** Characteristics of the case–control studies included.

Author, Year	Country	Sample Size		Sample Type	Staging	Diagnostic Method	Phase of Menstrual Cycle	Method of miRNA Detection	Method of Exosome Detection	Method of Exosome Isolation
		Case	Control							
Harp et al., 2016 [23]	USA	5	5	Eutopic endometrial tissue	-	Laparoscopy	Secretory	RT-qPCR	1. Nanoparticle tracking 2. Transmission electron microscopy	-
Huang et al., 2024 [24]	China	10	10	Serum	III–IV	Laparoscopy	Proliferative	microarray	1. Nanoparticle tracking 2. Transmission electron microscopy 3. Western blotting	Ultracentrifugation
Jiang et al., 2022 [25]	China	25	25	1. Eutopic endometrial tissue 2. Uterine aspirate fluid	III–IV	Laparoscopy	-	RT-qPCR	1. Nanoparticle tracking 2. Transmission electron microscopy 3. Western blotting	Exosome Isolation Kit (Echobiotech, Beijing, China)
Wu et al., 2018 [26]	China	6	6	Ectopic endometrial tissue	III–IV	Laparoscopy	-	qRT-PCR	1. Transmission electron microscopy 2. Western blotting	Ultracentrifugation
Wu et al., 2022 [27]	China	42	24	Serum	I–IV	Laparoscopy	Proliferative secretory	1. Microarray 2. RT-PCR	1. Transmission electron microscopy 2. Western blotting	Ultracentrifugation
Zhang et al., 2020 [28]	China	20	20	Ectopic endometrial tissue	-	Laparoscopy + Histopathologic examination	-	1. Microarray 2. qRT-PCR	1. Nanoparticle tracking 2. Transmission electron microscopy 3. Western blotting	Differential centrifugation
Zhang et al., 2020 [29]	China	20	20	Serum	I–IV	Laparoscopy + Histopathologic examination	-	1. Microarray 2. qRT-PCR	1. Nanoparticle tracking 2. Transmission electron microscopy 3. Western blotting	Ultracentrifugation
Zhang et al., 2020 [30]	China	20	20	Ectopic endometrial tissue	III	Laparoscopy	Proliferative	qRT-PCR	1. Transmission electron microscopy 2. Western blotting	SBI ExoQuick-TC
Zhang et al., 2022 [31]	China	24	20	1. Ectopic endometrial tissue 2. Eutopic endometrial tissue	-	Laparoscopy	Proliferative	1. Microarray 2. qRT-PCR	1. Nanoparticle tracking 2. Transmission electron microscopy 3. Western blotting	Ultracentrifugation
Zhang et al., 2023 [32]	China	5	5	Tubal fluid	-	MRI + Laparoscopy + Histopathologic examination	Secretory	1. Microarray 2. qRT-PCR	1. Nanoparticle tracking 2. Transmission electron microscopy 3. Western blotting	Ultracentrifugation
Zheng et al., 2023 [33]	China	11	11	1. Ectopic endometrial tissue 2. Leukorrhea	-	Laparoscopy + Histopathological examination	-	1. Microarray 2. RT-qPCR	1. Transmission electron microscopy 2. Western blotting	Ultracentrifugation
Zhou et al., 2020 [34]	China	3	3	Eutopic endometrial tissue	II–IV	Laparoscopy	Secretory	qRT-PCR	Transmission electron microscopy	ExoQuick-TC Exosome Precipitation Solution (System Biosciences)

MRI: magnetic resonance imaging; qRT-PCR: quantitative reverse transcription polymerase chain reaction.

**Table 2 ijms-26-04564-t002:** Quality assessment of included research papers using the Newcastle–Ottawa Scale for case–control studies.

Reference	Selection				Comparability	Exposure			Overall Quality Score
	An Adequate Definition of Cases	Representativeness of Cases	Selection of Controls	Definition of Controls	Comparability of Cases and Controls Based on Design or Analysis	Ascertainment of Exposure	Same Method for Ascertainment of Cases and Controls	Non-Response Rate	
Harp et al., 2016 [23]	*	-	*	*	-	*	*	-	5/9
Huang et al., 2024 [24]	*	*	*	*	-	*	*	-	6/9
Jiang et al., 2022 [25]	*	*	*	*	-	*	*	-	6/9
Wu et al., 2018 [26]	*	*	-	*	**	*	*	-	7/9
Wu et al., 2022 [27]	*	*	*	*	**	*	*	-	8/9
Zhang et al., 2020 [28]	*	*	*	*	-	*	*	-	6/9
Zhang et al., 2020 [29]	*	*	-	*	-	*	*	-	5/9
Zhang et al., 2020 [30]	*	*	*	-	-	-	*	-	4/9
Zhang et al., 2022 [31]	*	*	*	*	**	*	*	-	8/9
Zhang et al., 2023 [32]	*	*	-	-	-	*	*	-	4/9
Zheng et al., 2023 [33]	*	*	*	*	-	*	*	-	6/9
Zhou et al., 2020 [34]	*	*	*	-	-	*	*	-	5/9

**Table 3 ijms-26-04564-t003:** Significantly dysregulated exosomal microRNAs in endometriosis.

Reference	Number of Differential microRNA	Differential miRNA Expression Criteria	Dysregulated Exosomal microRNA
**Tubal fluid**			
Zhang et al., 2023 [32]	14	Fold change >2 *p* < 0.05	Upregulated (↑): miR-6087; miR-4443; miR-5194; miR-6834-3p. Downregulated (↓): miR-6747-5p; miR-1273f; miR-5699-5p; **miR-10b-3p**; miR-3911; miR-4419a; miR-4441; miR-4655-3p; miR-6778-5p; miR-6845-5p.
**Serum**			
Huang et al., 2024 [24]	50	Fold change >0.5 *p* < 0.05	Upregulated (↑): miR-4689; miR-4651; miR-6086; miR-6836; miR-551b-5p; miR-3124-5p; miR-671-5p. Downregulated (↓): miR-4497; miR-6779-5p; miR-185-5p; let-7i-5p; miR-27b-3p; **miR-22-3p**; miR-19b-3p; miR-221-3p; miR-3135b; miR-18a-5p; miR-423-3p; miR-27a-3p; miR-4454; miR-151a-3p; **miR-1273g-3p**; miR-4429; miR-423-5p; miR-320d; miR-191-5p; miR-151a-5p; miR-23b-3p; miR-24-3p; miR-17-5p; miR-107; miR-103a-3p; miR-320c; **miR-320a**; **miR-320b**; miR-199a-3p; miR-199b-3p; miR-20a-5p; miR-26a-5p; miR-23a-3p; miR-139-5p; miR-93-5p; miR-361-5p; let-7g-5p; let-7f-5p; miR-584-5p; miR-223-3p; miR-151b; let-7e-5p; miR-25-3p.
Wu et al., 2022 [27]	45	Fold change >2 *p* < 0.05	Upregulated (↑): miR-6795-3p; miR-6889-3p; miR-4731-3p; miR-6731-3p; miR-6760-3p; miR-6870-3p; miR-7114-3p; miR-424-5p; miR-6813-3p; miR-3940-3p; miR-1271-5p; miR-1303; miR-6785-3p; miR-3675-3p; **miR-1273g-3p**; miR-3180-5p; miR-4475; miR-146b-3p; miR-500a-3p; miR-877-5p; miR-885-3p; miR-6818-3p; miR-6751-5p; miR-539-5p; miR-32-3p; miR-4505. Downregulated (↓): miR-128-1-5p; miR-215-5p; miR-26b-5p; miR-4453; miR-510-3p; miR-3140-3p; miR-3929; miR-3678-3p; miR-4303; miR-6743-3p; miR-514a-3p; miR-4315; miR-3074-5p; miR-628-3p; miR-6836-5p; miR-659-5p; miR-323b-5p; miR-5091; miR-3910.
Zhang et al., 2020 [29]	24	Fold change >1 *p* < 0.05	Upregulation (↑): miR-197-5p; **miR-22-3p**; **miR-320a**; **miR-320b**; miR-3692-5p; miR-4476; miR-4530; miR-4532; miR-4721; miR-4758-5p; miR-494-3p; miR-6126; miR-6734-5p; miR-6776-5p; miR-6780b-5p; miR-6785-5p; miR-6791-5p; miR-939-5p. Downregulated (↓): miR-134-5p; miR-3141; miR-4499; miR-6088; miR-6165; miR-6728-5p.
**Endometrial tissue**			
Harp et al., 2016 [23]	-	*p* < 0.05	Upregulation (↑): miR-21
Jiang et al., 2022 [25]	21	Fold change ≥1.50 *p* < 0.05	Upregulation (↑): miR-210-3p; miR-30d-3p; miR-141-5p; **miR-200c-3p**; miR-224-5p; miR-4521; miR-29b-3p; miR-30b-5p; miR-16-2-3p; miR-345-5p; miR-375-3p; miR-9-3p; miR-9-5p; miR-190b-5p; miR-34c-5p. Downregulation (↓): miR-708-5p; miR-143-5p; miR-132-3p.
Wu et al., 2018 [26]	1	-	Downregulation (↓): miR-214.
Zhang et al., 2020 [28]	20	Fold change >1.5 *p* < 0.05	Upregulation (↑): **miR-22-3p**; miR-28-5p; miR-302a; **miR-320b**; miR-3118; miR-3168; **miR-425-5p**; miR-4256; miR-4447; miR-507; miR-596; miR-5582; miR-610; miR-663a; miR-6720; miR-133. Downregulation (↓): miR-296; miR-1912; miR-2113; miR-3188.
Zhang et al., 2020 [30]	-	*p* < 0.05	Downregulation (↓): miR-214-3p.
Zhang et al., 2022 [31]	-	*p* < 0.05	Downregulation (↓): miR-30c.
Zheng et al., 2023 [33]	217	*p* < 0.05	Upregulation (↑): miR-202-3p; miR-202-5p.
Zhou et al., 2020 [34]	49	Fold change >1.5 *p* < 0.05	Upregulation (↑): **miR-10b-3p**; miR-1468-5p; miR-125b-2-3p; miR-2682-5p; miR-494-5p; **miR-200c-3p**; miR-345-3p; miR-450a-2-3p; miR-3180; miR-615-3p; miR-196a-5p; miR-6873-3p; miR-4483; miR-3661; miR-379-3p; miR-411-3p; miR-142-3p; miR-4532; miR-6131; miR-3195; let-7c-3p; miR-1343-3p; miR-1299; miR-99a-5p; let-7c-5p; miR-10b-5p. Downregulation (↓): **miR-425-5p**; miR-4671-3p; miR-4664-3p; miR-4473; miR-6500-3p; miR-3653-3p; miR-4421; miR-378h; miR-29c-5p; miR-4262; miR-1269a; miR-124-3p; miR-1273h-3p; miR-1273h-5p; miR-548am-3p; miR-548f-3p; miR-6859-5p; miR-548j-3p; miR-4467; miR-4648; miR-365a-3p; miR-365b-3p; miR-486-3p.
**Uterine aspirate fluid**			
Jiang et al., 2022 [25]	9	Fold change ≥1.50 *p* < 0.05	Upregulation (↑): miR-210-3p; miR-20b-5p; miR-625-5p; miR-342-5p; miR-155-5p; miR-146a-5p; miR-130b-3p. Downregulation (↓): miR-335-3p; miR-132-5p.
**Leukorrhea**			
Zheng et al., 2023 [33]	217	*p* < 0.05	Upregulation (↑): miR-202-3p; miR-202-5p.

Words in bold represent the microRNAs reported in more than one study.

## Data Availability

The datasets generated during and/or analyzed during the current study are available from the corresponding author upon reasonable request.

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
