# Peer review of "Exosomal MicroRNAs as Epigenetic Biomarkers for Endometriosis: A Systematic Review and Bioinformatics Analysis"

_ijms, 2025, doi:10.3390/ijms26104564_

Round 1
Reviewer 1 Report
Comments and Suggestions for Authors
In the manuscript“Exosomal microRNAs as epigenetic biomarker for endometrio-
sis: A Systematic Review and Bioinformatics Analysis”, the author describes the potential of exosomal miRNAs as biomarkers in endometriosis, and identified several microRNAs that were consistently reported across multiple studies.
My comments on this manuscript:
1. The size of exosome in the description “These are small, membrane-bound vesicles (30–100 nm in diameter) secreted by various cell types. ” should be carefully check out. Other studies report the diameter of exosome is between 30 and 150nm.
2. This article discribes the existing research results from relevant literatures. However, the authors did not clearly point out the unsolved problems and the future directions in the field.
3. At the end of the paper, GO and KEGG bioinformatics analysis were carried out on the related microRNAs, the list of target genes should be added to support this description.
4. The references in the sentence of “even studies did not specify the stage of endometriosis 101 [23,24,25,28,31-33].” Should correct to “even studies did not specify the stage of endometriosis 101 [23-25,28,31-33].”
Author Response
Comment 1: The size of exosome in the description “These are small, membrane-bound vesicles (30–100 nm in diameter) secreted by various cell types. ” should be carefully check out. Other studies report the diameter of exosome is between 30 and 150nm.
Response 1: Thank you for the feedback. We agree and have therefore corrected it to 30-150 nm.
Comment 2: This article discribes the existing research results from relevant literatures. However, the authors did not clearly point out the unsolved problems and the future directions in the field.
Response 2: We understand the concern; however, it was stated in the "Conclusion" section that further studies are necessary, as follows: “Ultimately, while exosomal miRNAs offer a window into the complex pathophysi-ology of endometriosis, large-scale, standardized studies are essential to confirm these findings and enable the use of miRNAs in early diagnosis and targeted treatments for en-dometriosis”. Also, in the “Discussion” section, as follows: “Finally, although we identified promising exosomal miRNAs, the validation of these in independent large-scale cohorts is necessary to confirm their diagnostic accuracy and clinical utility”.
Comment 3: At the end of the paper, GO and KEGG bioinformatics analysis were carried out on the related microRNAs, the list of target genes should be added to support this description.
Response 3: We agree with the suggestion and would like to inform you that we have added the table containing the target genes as supplementary material, which is referenced in the "4.7. Bioinformatic Analysis" subsection as follows: "The list of target genes is present in Supplementary Material 3".
Comment 4: The references in the sentence of “even studies did not specify the stage of endometriosis 101 [23,24,25,28,31-33].” Should correct to “even studies did not specify the stage of endometriosis 101 [23-25,28,31-33].”
Response 4: Thank you for your observation, and we would like to inform you that this adjustment has been made.

Reviewer 2 Report
Comments and Suggestions for Authors
Authors presented an iteresting paper regarding miRNAs as biomarkers of endometriosis. The value of the paper is highlighted by bioinformatics, that could match clinical findings with predictes outcomes. I have few minor points I would like to address to the Authors:
1. There are miRNAs like to be biomarkers. In my opinion it would be interesting if Authors could provide any mesaureable data for the miRNAs, such as AUC, OR ot other paramters.
2. Authors discussed exosomal miRNAs. What is a predominance of the exosomal miRNAs over the free circulating miRNAs?
3. Please more carefully discuss what is a major issue for the biomarkers selection.
Author Response
Comment 1: There are miRNAs like to be biomarkers. In my opinion it would be interesting if Authors could provide any mesaureable data for the miRNAs, such as AUC, OR ot other paramters.
Response 1: Thank you for your question. We did not aim to determine these calculations for diagnostic accuracy.
Comment 2: Authors discussed exosomal miRNAs. What is a predominance of the exosomal miRNAs over the free circulating miRNAs?
Response 2: We understand the concern; therefore, we have added the following in the first paragraph of the discussion: “Exosomal miRNAs, protected within extracellular vesicles, exhibit greater stability and targeted delivery compared to free circulating miRNAs. This enhanced stability and specificity contribute to their potential predominance in various biological contexts [35]”.
Comment 3: Please more carefully discuss what is a major issue for the biomarkers selection.
Response 3: We understand the concern; however, the rationale for using exosomal miRNAs as biomarkers lies in the fact that circulating miRNAs are more unstable due to RNase activity, as mentioned in the last paragraph of the introduction, as follows: "However, circulating miRNAs in the bloodstream are vulnerable to degradation by endogenous RNases. In contrast, exosomal miRNAs are stable in the blood and generally resistant to ribonuclease activity, (...)".

Round 2
Reviewer 1 Report
Comments and Suggestions for Authors The manuscript has been sufficiently improved.
Reviewer 2 Report
Comments and Suggestions for Authors
Auhors replied to the all addressed queries. I have no additional comments.